# Effect of Different Extraction Methods on Physicochemical Characteristics and Antioxidant Activity of C-Phycocyanin from Dry Biomass of *Arthrospira platensis*

**DOI:** 10.3390/foods11091296

**Published:** 2022-04-29

**Authors:** Qian Chen, Shuhui Li, Hua Xiong, Qiang Zhao

**Affiliations:** 1State Key Laboratory of Food Science and Technology, Nanchang University, Nanchang 330047, China; ncuspychenqian@163.com (Q.C.); huaxiong100@126.com (H.X.); 2Jiangxi Academy of Agricultural Sciences, Nanchang 330200, China; linlish1023@163.com

**Keywords:** C-phycocyanin, high-pressure cell disruption, antioxidant activity, extraction method, *Arthrospira platensis*

## Abstract

The effect of four different extraction methods on physicochemical characteristics and functionalities of chloro-phycocyanin (CP) was investigated. Swelling (S-CP), freezing and thawing (4FT-CP), ultrasonication with freezing and thawing (4FT+U-CP), and the high-pressure cell disruption (HPCD-CP) process affected CP differently, thus resulting in different levels of solubility, DPPH scavenging activity, ABTS scavenging activity, and reducing power. Among the four CPs, HPCD-CP had the highest CP content (15.3%), purity (1.66 ± 0.16), and ∆E value but the lowest ∆b value. The ζ potential of HPCD-CP (−38.8 mV) was the highest, but the average particle size of 4FT+U-CP (719.1 nm) was the highest. UV-Vis absorption spectra and fluorescence spectra illustrated that high-pressure cell disruption-assisted extraction had more profound impacts on the microenvironment of tetrapyrrole chromophores, the environment of aromatic amino acids, and the phycocyanobilin of CP. Furthermore, HPCD-CP and 4FT-CP showed higher solubility and antioxidant activities than S-CP, especially 4FT+U-CP. The results obtained in this study demonstrate that HPCD technology could obtain a food-grade C-phycocyanin product with higher CP concentration, purity, solubility, and antioxidant activity.

## 1. Introduction

Spirulina (*Arthrospira platensis*) is a blue-green alga rich in protein (60–70%, including phycobiliproteins). It is a supplement containing various kinds of essential amino acids, vitamins, minerals, chlorophyll, carotenoids, ascorbic acid, and phenolic compounds, thus playing a significant role in scavenging free radicals and preventing oxidative stress-related diseases [1,2,3,4]. Phycobiliproteins (PBPs) are composed of proteins and phycobilins and are covalently bound with open-chain tetrapyrrole chromophores of phycobilins via the cysteine amino acid of proteins [5,6]. Chloro-Phycocyanins (CP, blue, λ_max_ = 610–620 nm), phycoerythrins (PE, pink-purple, λ_max_ = 540–570 nm), allophycocyanins (APC, bluish-green, λ_max_ = 650–655 nm), and phycoerythrocyanins (PEC, orange, λ_max_ = 560–600 nm) are the four subclasses of PBPs. Commonly used as a blue pigment, CP constitutes the main part of PBPs in *Arthrospira platensis* when compared with PE, PEC, and APC, and it can be applied in many ways, such as in natural pigments for food and cosmetics and fluorescent tags in biomedical research [7]. In addition, owing to its high antioxidant activity against hydroxyl radicals [8] and free-radicals [9], it has been used as an agent anti-inflammatory and potential therapeutic agent for oxidative stress-induced diseases [2].

Nowadays, there are many studies aimed at CP extraction from Spirulina with different methods of cell disruption, such as thawing and homogenizing [10], freezing and thawing [11], supercritical CO_2_ extraction followed by the electrocoagulation method [12], high-pressure cell disruption [13], and ultrasound-assisted extraction [14]. Chittapun, Jonjaroen, and Charoenrat [15] indicated that the freezing and thawing technique showed better performance in terms of CP concentration than the pulsed electric field technique, while CP obtained from the pulsed electric field technique showed a higher purity. Li et al. [16] investigated the high-pressure process, pulsed electric field, and ultrasonication process techniques and concluded that most phycocyanins could be released from the broken small particles by ultrasonic waves and pass through the cell walls of Spirulina. It is worth noting that high-pressure cell disruption was superior to bead-beating for the extraction of CP [17]. Many of the reported processes to date are expounded in terms of recovery yield, purity indices, as well as the yield of CP from different primary extraction methods. However, as far as we know, there are few studies on the influences of different extraction methods of CPs on their functional and physicochemical properties [18]. 

Hence, in this work, we investigated and compared the effects of four extraction methods, namely the swelling process (stirred continuously at 25 °C for 12 h, S-CP), freezing and thawing process (freezing at −20 °C for 12 h and thawing to 27 °C for 12 h, 4 cycles, 4FT-CP), freezing and thawing combined with ultrasonication process (ultrasonication at 40% for 30 min after freezing and thawing process, 4FT+U-CP), and high-pressure cell disruption process (70 MPa for 1 cycle, HPCD-CP), on the physicochemical characteristics and antioxidation of CP from dry biomass of *Arthrospira platensis*. In this study, high-pressure cell disruption (HPCD) was developed as a new method using a mechanical process in the production of C-phycocyanin from the dry biomass of *Arthrospira platensis*. The findings of this study also provide more evidence to support further evaluation of the use of mechanical extraction (for other microalgae protein) for future applications in the functional food formulation industry.

## 2. Materials and Methods

### 2.1. Materials

Dry biomass of *Arthrospira platensis* was supplied by Jiangxi Zhongzao Biotechnology Co., Ltd (Ruijin, China). and stored at −20 °C. Petroleum ether (boiling range was 30~60 °C) and ammonium sulfate ((NH_4_)_2_SO_4_) were bought from Xilong Scientific Co., Ltd. (Guangdong, China). Two, 2′-azinobis-(3-ethylbenzothiazoline-6-sulfonic acid) (ABTS) and 1,1-diphenyl-2-picrylhydrazyl (DPPH) were purchased from Sigma Chemical Co. (St. Louis, MO, USA). All the reagents applied were of analytical grade.

### 2.2. Primary Extraction of C-Phycocyanin

#### 2.2.1. Pretreatment of *Arthrospira platensis*


Dry biomass of *Arthrospira platensis* was defatted three times with petroleum ether (60–90 °C) at a ratio of 1:10 (*w*/*v*). After decanting the supernatant, the Arthrospira platensis biomass was air-dried in a fume hood for two days, so the residual petroleum ether was allowed to evaporate.

#### 2.2.2. Swelling Extraction Process

The swelling extraction process was performed based on the method described by Li et al. [16] with slight modifications. Predetermined weights of defatted *Arthrospira platensis* were dispersed in distilled water at a ratio of 1:20 (S/L) and stirred continuously for 12 h at room temperature in the dark. The supernatant and pellet were separated after being centrifuged in a centrifuge (LXJ-IIB, Anting Scientific Instrument Factory, Shanghai, China) at 4500 rpm for 30 min. Lastly, the supernatant was kept for purification, named S-CP.

#### 2.2.3. High-Pressure Cell Disruption 

The high-pressure cell disruption extraction process was performed by the method of Drévillon et al. [19]. Predetermined weights of defatted *Arthrospira platensis* were dispersed in distilled water at a ratio of 1:20 (S/L) and stirred continuously for 4 h at room temperature in the dark for pre-soaking. Then, the slurry was subjected to high-pressure cell disruption with a high-pressure cell disruptor (TS-series 1.1 kW model, Constant Systems Ltd., South Easton, UK) at 70 MPa for 1 cycle. Lastly, the supernatant, named HPCD-CP, was obtained by centrifugation and kept for purification.

#### 2.2.4. Freezing and Thawing 

The freezing and thawing extraction process was accomplished according to Antecka et al. [20]. After pre-soaking in distilled water, the predetermined weights of defatted *Arthrospira platensis* were dispersed at a ratio of 1:20 (S/L). Then, the slurry underwent 4 freeze-thaw cycles. In each cycle, the slurry was continuously frozen at a certain temperature of −20 ± 2 °C for 12 h, followed by thawing for 12 h at room temperature. Lastly, the supernatant, named 4FT-CP, was obtained by centrifuge and kept for purification.

#### 2.2.5. Ultrasonication with Freezing and Thawing 

Ultrasonication with the freezing and thawing extraction process was accomplished based on the work performed by Tavanandi et al. [21]. Predetermined weights of degreased *Arthrospira platensis* were dispersed in distilled water at a ratio of 1:20 (S/L) for pre-soaking. Then, referring to our previous pre-experiment, the slurry was subjected to ultrasonication at an amplitude of 40% for 30 min (KQ-800KDE, Kunshan Ultrasonic Instrument Co., Ltd., Suzhou, China) after 4 freeze-thaw cycles. Lastly, the supernatant, named 4FT+U-CP, was obtained by centrifugation and kept for purification.

### 2.3. Purification of CP

The purification of CP was accomplished based on the method of Patel et al. [22] with minor modifications. The crude CP from *Arthrospira platensis* was enriched first by salting it out with solid ammonium sulfate at 25% (*w*/*v*) for 4 h. The slurry was centrifuged at 4500 rpm for 30 min, then the supernatants were recovered and further precipitated by adding solid ammonium sulfate to 50% (*w*/*v*) saturation and allowing the slurry to stand overnight at 4 °C. The precipitated proteins, containing mainly CP, were collected by centrifugation at 4500 rpm for 30 min. The isolated protein was re-suspended in distilled water and then centrifuged again at 8000 rpm for 20 min (Hitachi high-speed refrigerated centrifuge, Himac CR21N, Tokyo, Japan). Lastly, the supernatants were dialyzed (8000–12,000 Da) at 4 °C to remove the solid ammonium and freeze-dried (LGJ-18 Vacuum freeze dryer, Songyuanhuaxing Technology Develop Co., Ltd., Beijing, China) for storage. The protein content of CP was determined using the Kjeldahl method. 

### 2.4. Color Measurement

A CIE-Lab color scale was used for measuring the spent biomass of CP by a colorimeter (HP-2136 Portable Colorimeter, Puxi Shanghai, China), according to the method of Tavanandi et al. [21]. The sample solution was placed in a 20 mL glass colorimetric bottle with distilled water as a standard [23]. All these processes were performed on white A4 paper three times. The brightness of the color was denoted by L*, where the number “0” represented black and “100” represented white. The +a* value indicated that the sample was red, while −a* was used to indicate green. +b* indicated that the sample color was yellow, while −b* indicated blue [24]. The ∆E*, ∆L*, ∆a*, and ∆b* values were read directly from the instrument.

### 2.5. UV-Vis Spectra 

The absorption spectra of CP were measured on a UV-Vis spectrophotometer (TU-1900, Puxi General Instrument Co., Ltd., Beijing, China). The spectral scanning was performed in a wavelength range from 250 nm to 700 nm. The absorbance at λ = 620 nm, 652 nm, and 280 nm was extracted to calculate the CP, APC, and total protein concentration. Bennett and Bogorad [25] determined the CP concentration via the following equation:(1)$$\mathrm{CP}\;\left(\mathrm{mg}/\mathrm{mL}\right)=\left[A_{620}-0{.474(A}_{652})\right]/5.34$$

The purity of CP was expressed as P (purity) and calculated as follows:(2)$$P=\frac{A_{620}}{A_{280}}$$
where A_620_ represents the absorption of phycocyanin, while A_280_ represents the absorption of total protein. 

The CP content of the samples and the yield were evaluated using the following equations (sample concentration was 1 mg/mL):(3)$$\mathrm{CP}\;\mathrm{content}\;\left(\%\right)=\left[\frac{\mathrm{CP}\;(\mathrm{mg}/\mathrm{mL})}{\mathrm{Sample}\;\mathrm{content}\;(\mathrm{mg}/\mathrm{mL})}\right]\times 100$$
(4)$$\;\mathrm{Yield}\;\left(\%\right)=\frac{\mathrm{Dry}\;\mathrm{CP}\;\mathrm{powder}\;\mathrm{from}\;\mathrm{primary}\;\mathrm{extraction}\;\mathrm{methods}\;(g)\;}{\mathrm{Dry}\;\mathrm{biomass}\;\mathrm{of}\;Arthrospira\;platensis\;(g)}\times 100\%\;$$

### 2.6. Fluorescence Spectra 

The intrinsic fluorescence of the samples was measured by a fluorescence spectrophotometer (F-7000, Hitachi, Kyoto, Japan) at a concentration of 1 mg/mL. When the sample was excited at 280 nm and 580 nm, the emission band was recorded. We set a 5 nm slit for emission.

### 2.7. Fourier Transform Infrared Spectroscopy (FTIR) 

The infrared spectra of freeze-dried CP were measured with KBr pellets on an FTIR spectrophotometer (Nicolet 5700, Thermo Fisher, Boston, MA, USA). The spectra were scanned with a wavenumber range between 400–4000 cm^−1^, at a resolution of 4 cm^−1^.

### 2.8. Thermogravimetric Analysis (TGA)

The measurements were performed according to Lemos et al. [26] via a pre-calibrated Perkin Elmer thermogravimetric analyzer (TGA 4000, Perkin Elmer, Waltham, MA, USA). Conditions for the use of a platinum crucible were as follows: an approximately 5 mg sample mass was heated from 30 °C to 800 °C at a heating rate of 10 °C/min with a nitrogen flow rate of about 40 mL/min.

### 2.9. ζ Potential and Particle Size

The *ζ* potential and particle size distribution of the samples were detected by a Nano ZS90 Malvern Zetasizer (Malvern Instrument, Malvern, Worcestershire, UK). The samples were dispersed in distilled water at a concentration of 1 mg/mL. Each measurement was performed three times. 

### 2.10. Protein Solubility

The protein solubility measurement was conducted according to the method of Bera and Mukherjee [27] with some modifications. To better determine the impact of the pH on the functional properties of CP, the pH of CP dispersion in distilled water (l mg/mL) was increased from 2.0 to 12.0 (intervals of 1.0). The sample solutions were stirred at room temperature for 2 h and then centrifuged at 8500 rpm for 10 min. The content of the protein in the supernatant was measured using a bicinchoninic acid (BCA) protein quantification assay (Thermo Fisher Scientific, Darmstadt, Germany). The protein solubility (%) was calculated by the following equation:(5)$$\mathrm{Solubility}\;\left(\%\right)=\frac{\mathrm{Protein}\;\mathrm{concentration}\;\mathrm{in}\;\mathrm{the}\;\mathrm{supernatant}\;\left(\mathrm{mg}/\mathrm{mL}\right)}{\mathrm{Sample}\;\mathrm{concentration}\;\left(\mathrm{mg}/\mathrm{mL}\right)}\times 100\%$$

### 2.11. Antioxidant Activity

#### 2.11.1. DPPH Scavenging Activity 

The DPPH radical scavenging activity was determined using the previously reported method [9]. An amount of 100 µL of distilled water, with 0.5, 1.0, 1.5, 2.0, 2.5, and 3.0 mg of the sample solutions, was placed in a 96-well plate and then mixed with a 100 µL of DPPH ethanol solution (0.1 mmol/L). The sample containing a DPPH solution without the sample served as a control. A blank sample containing a sample solution with ethanol was also prepared. The mixture was incubated at 37 °C for 30 min and measured at 517 nm. The DPPH scavenging activity of the sample was evaluated using the following equation:(6)$$\mathrm{DPPH}\;\mathrm{radical}\;\mathrm{scavenging}\;\mathrm{activity}\;\left(\%\right)=\frac{A_{\mathrm{control}}-A_{\mathrm{sample}}{+A}_{\mathrm{blank}}}{A_{\mathrm{control}}}\times 100\%$$
where A_control_ indicates the absorbance control group; A_sample_ indicates sample absorbance rate; A_blank_ indicates blank absorbance.

#### 2.11.2. ABTS Scavenging Activity 

The ABTS radical scavenging activity was assessed by the inhibition percentage of the ABTS radical, as described by Wang et al. [28]. 200 µL of diluted ABTS radical solution was blended with 10 µL of 0.5, 1.0, 1.5, 2.0, 2.5 and 3.0 mg/mL of sample solutions. After 6 min incubation at 37 °C, the absorbance against the corresponding blank was measured at 734 nm. The solution containing ABTS solution without sample served as a control. The ABTS scavenging activities of the samples were evaluated using the following equation:(7)$$\mathrm{ABTS}\;\mathrm{scavenging}\;\mathrm{activity}=\frac{A_{\mathrm{control}}-A_{\mathrm{sample}}{+A}_{\mathrm{blank}}}{A_{\mathrm{control}}}\times 100\%$$
where A_control_ and A_sample_ stand for the absorbance without/with sample, respectively; A_blank_ indicates the absorbance of blank group. 

#### 2.11.3. Reducing Power 

The reductive capacity was evaluated by the method previously reported by Liu et al. [9] with slight modifications. Two milliliters of phosphate-buffered saline (PBS, pH 6.6, 0.2 mol/L) and one milliliter of a potassium ferricyanide solution (1%, *w*/*v*) were added to one milliliter of 0.5, 1.0, 1.5, 2.0, 2.5, and 3.0 mg/mL of the sample solution and incubated at 50 °C for 20 min. Then, 2 mL of a trichloroacetic acid solution (10%, *w*/*v*) was added to the mixture to terminate the reaction. An amount of 2 mL of distilled water and a 0.5 mL ferric chloride solution (0.1%, *w*/*v*) were added to the mixture and reacted for 10 min. The absorbance was recorded at 700 nm against a blank containing all reagents except the sample. 

### 2.12. Statistical Analysis

All assays were performed on three samples, with the results recorded as a mean ± standard deviation, and the significant differences (*p* < 0.05) of data were processed using Tukey’s test by analysis of variance (ANOVA) from Origin 2018 software (OriginLab Corporation, Northampton, MA, USA).

## 3. Results and Discussion

### 3.1. Yield, Purity, and Color of CP

Table 1 shows the CP content, protein content, and yield. The CP yields were 15.9% for S-CP, 5.80% for HPCD-CP, 9.80% for 4FT-CP, and 15.92% for 4FT+U-CP. According to the result, the 4FT+U-CP had the highest efficiency of extraction, while the HPCD-CP presented the lowest CP sample yield; this was consistent with the study of Tavanandi et al. [21], as ultrasonication presented a stronger ability to break down the *Arthrospira platensis* cell walls [16]. During ultrasonication, intense local shock waves, corresponding to thousands of atmosphere pressure, were produced to destroy the cell walls. In Figure 1A, three peaks at 350 nm, 375 nm, and 424 nm were also present in the spectra of the ultrasonicated sample; this is because more of the other compositions were obtained in crude 4FT+U-CP. In addition, the extraction methods also had a certain effect on the CP content of the samples (*p* < 0.05). HPCD-CP had the highest CP content (15.3%) but the shortest extraction time compared to S, 4FT, and 4FT+U. The results of Li et al. [16] indicated that most CPs can be obtained in about 3 h, so a longer extraction time may cause the leakage of excessive substances from the Spirulina cells. For CP, the ratio of the active substance to the total quantity of protein (A_620_/A_280_) is defined as purity. Purity 0.7 is considered food-grade, 3.9 reactive-grade, and above 4.0 analytical-grade [29]. As shown in Figure 1B, the purities of all samples were over 0.7, so, in the case of foods, these extraction methods, with a two-step solid ammonium sulfate purification process, were proved to be industrially applicable. The purity of HPCD-CP (1.66 ± 0.16) and the ∆E (Figure 1C) were the highest, while the ∆b value was the lowest, which means that the color of HPCD-CP is significantly bright blue (*p* < 0.05). At the same time, the color of HPCD-CP also proved that both the purity and CP content results of HPCD-CP were the highest. 

### 3.2. ζ Potential and Particle Size

The ***ζ*** potential of the CPs extracted from different methods are shown in Figure 1D. According to the figure, the *ζ* potential of the CPs decreased in the following order: HPCD-CP (−38.8 mV) > 4FT+U-CP (−41.1 mV) > 4FT-CP (−46.1 mV) > S-CP (−52.0 mV). The average particle size of the CPs extracted from different methods are presented in Figure 2A in the following decreasing order: 4FT+U-CP (719.1 nm) > HPCD-CP (654.2 nm) > 4FT-CP (536.3 nm) > S-CP (451.4 nm). An increase in average particle size was observed with a further increase in the ***ζ*** potential; this can be interpreted as if a sample has a high ***ζ*** potential (either positive or negative), providing further away from the zero point so it can be electrically stable. When the ***ζ*** potential is low, a sample solution will be unstable (tend to coagulate or flocculate easily) [30]. S-CP had the lowest ***ζ*** potential and particle size, significantly different (*p* < 0.05) from those of 4FT+U-CP and HPCD-CP. It can be suggested that extraction with mechanization may result in an increase in the protein particle size, especially under low intensities [31,32].

### 3.3. Spectrophotometric of CP

Figure 1A shows the UV-visible absorption spectra of the CPs. The CPs displayed three relatively strong absorption peaks located at 280 nm, 350 nm, and 620 nm, similar to the maximum absorption of aromatic amino acids, denatured phycocyanin, or the unbound phycocyanobilin chromophore in its cyclic conformation and protein–pigment complex, respectively [22]. The absorption intensity, at 620 nm, increased in the following order: S-CP < 4FT+U-CP < 4FT-CP < HPCD-CP; this is similar to the previous results of the CP content. According to the intensity at 350 nm of the samples, S-CP was lower than 4FT-CP and 4FT+U-CP but higher than HPCD-CP, indicating that the microenvironments of the tetrapyrrole chromophores of 4FT-CP and 4FT+U-CP are more hydrophobic, followed by S-CP. The increased intensity at 350 nm (compared to HPCD-CP) might be related to the fact that there are more denatured phycocyanin or unbound phycocyanobilin chromophore caused by a longer extraction time of S-CP, 4FT-CP, and 4FT+U-CP [16,22]. 

As shown in Figure 2B, there was a blue shift (compared to the emission maximum of the S-CP result) from 344 nm to 338 nm in the other three CPs upon the excitation of 280 nm, indicating that the microenvironment around the aromatic amino acids was more hydrophobic [33]. Zhou et al. [34] found that an ultrasound treatment on egg white protein increased the number of hydrophobicity groups. The HPCD-CP exhibited a strong emission peak at 669 nm upon excitation at 580 nm (Figure 2C), and there was a blue shift of the fluorescence emission wavelength in the other three CPs, indicating that the microenvironment of phycocyanobilin was more hydrophobic. Additionally, the largest decrease in fluorescence intensity was observed in HPCD-CP. The reason for the changes in the protein molecules might be the different production conditions of different methods, such as production time, process, and intensity.

Three typical protein bands of amide I (1600–1700 cm^−1^), amide II (1500–1580 cm^−1^), and amide III (1200–1400 cm^−1^) were observed by FTIR spectroscopy [35]. Shown in Figure 2D are the strong absorption peaks at 1648 cm^−1^, 1648 cm^−1^, 1658 cm^−1^, and 1649 cm^−1^, pertaining to S-CP, HPCD-CP, 4FT-CP, and 4FT+U-CP, respectively. Basically, C=O tensile (amide I), and the α-helix corresponds to 1660–1650 cm^−1^ separately. The absorption peaks of the 4FT-CP and 4FT+U-CP samples at 3400–3500 cm^−1^ had a blue shift, and the peak shape became wider, indicating that the amide carbonyl group vibrated along the protein backbone. Table 2 shows the estimation of the secondary structure in the amide I region of the CPs obtained by different methods. The content of the random coil was 22.46% for HPCD-CP but without the α-helix. The contents of the α-helix were 19.16%, 28.31%, and 28.66% for S-CP, 4FT-CP, and 4FT+U-CP, respectively. The structure of CPs may be disintegrated and transformed to an ordered structure during swelling, repeated freezing and thawing, and the ultrasonic extraction process. Thus, these findings confirm that different extraction methods have a significant effect on the protein secondary structure of CPs.

### 3.4. Thermogravimetric Analysis (TGA)

Figure 3 shows that the TGA pyrolysis characteristics of the CPs extracted by different methods can be divided into three stages. In the first stage, most of the adsorbed water and bound water began to become lost at temperatures of 55–150 °C [36]. The greatest reduction in the biomass was observed in the second stage, at temperatures of 150–600 °C, at which the main organic compounds of the microalgal biomass, such as lipids, proteins, and carbohydrates, decompose, which is why this stage is also known as the active pyrolysis zone. In the third stage, thermally stable compounds decomposed at 600–800 °C and formed biochar. The CPs’ TGA curves showed a biomass loss of 1–5% in the first stage, decomposition of 59–75% in the second stage, and 8–18% in the last stage. The microalgal biomass decomposition in the second stage was lower than that which was previously reported [37]. A TGA spectroscopic analysis of all CPs suggested that the devolatilization peak, at 311–325 °C, showed the maximum decomposition of CP. Another important peak, to the left of the main peak, was also observed at 271–282 °C. The first and second stage temperatures of S-CP were 70.8 °C and 325.9 °C; those of 4FT-CP, HPCD-CP, and 4FT+U-CP were 5.25 °C and 4.84°C, 10.76 °C and 14.72 °C, and 15.72 °C and 8.34 °C lower than S-CP, respectively. Temperature changes at all peaks showed differences in microalgal biomass, similar to the results of Pandey, Srivastava, and Kumar [38]. 

The temperatures at the three main phases of the S-CP and 4FT-CP samples were higher than those of the other two samples. Zhang et al. [39] reported that the ultrasonic treatment could disrupt the inherent structure and lead to poor thermal stability. Other work also indicated that water evaporation and weight loss can be influenced by the content of the protein [40]. Thus, the reason for the difference in the temperatures of the CPs may be determined by two factors: **(i)** the CPs obtained through high-pressure cell disruption and ultrasound, assisted by the freezing and thawing process, were extracted by mechanization, and **(ii)** the significantly different amount of the protein of these purified CP samples [41].

### 3.5. Solubility of CP 

The effect of different extraction methods and pH on protein solubility are shown in Figure 4A. These CPs from different extraction methods had the minimum solubility near pH 4 (<10%), and their solubility was higher at pH values below and above four, which corresponds to their isoelectric point [42]. Similar results have been found for casein, soybean meal, mung bean protein, and other microalgal proteins; for instance, Nannochloropsis oculate and Spirulina LEB 18, with minimal solubility near the isoelectric point of pH 4–5 [43,44]. The solubility of 4FT-CP increased in the range of pH 4–8, decreased in the range of pH 8–10, and had the maximum solubility (82.2 ± 1.6%) at pH 11. In addition, the solubility of 4FT-CP was higher than that of the other three CPs between pH 7 and pH 9 and at pH 11. Furthermore, its high solubility in the pH 6–9 range is very similar to that of common vegetable protein drinks, which is at about 6.8 to 7.0 [45], indicating that CPs with different extraction methods may be applicable in food processing. 

### 3.6. Antioxidant Activity of CPs

#### 3.6.1. DPPH Radical Scavenging Activities

The DPPH radical scavenging effects of the CPs extracted from distinct methods are shown in Figure 4B. The sample concentration that can inhibit 50% of free radicals was defined as IC_50_**.** The IC_50_ value of HPCD-CP (0.68 mg/mL) CP was slightly lower than those of 4FT-CP (0.79 mg/mL) and S-CP (0.71 mg/mL) but significantly lower than that of 4FT+U-CP (0.92 mg/mL). Obviously, we can see from the results that the CPs obtained through different extracted methods displayed an increasing radical scavenging activity with an increase in the CP purities, as previously reported [46]. When the concentration is at 1 mg/mL, the DPPH radical scavenging capacity of HPCD-CP (67.32%) was significantly higher than those of the other CP samples. 

#### 3.6.2. ABTS Radical Scavenging Activities

According to the analysis in Figure 4A, the IC_50_ values of the four CP samples are between 1.28 mg/mL and 1.51 mg/mL. The IC_50_ value of the 4FT+U-CP sample (1.51 mg/mL) was obviously higher than that of the other CP samples. It is concluded that ultrasound may destroy free radical scavenging components in biological systems [47]. Figure 4C shows the effect of the ABTS radical scavenging activity on the CPs. The ABTS radical scavenging ability of the four CP samples was similar when the concentration was at 0.5 mg/mL. The free radical scavenging activities of ABTS were 75.26% for S-CP, 76.01% for HPCD-CP, 74.62% for 4FT-CP, and 67.96% for 4FT+U-CP at a concentration of 3 mg/mL. In conclusion, the ABTS radical scavenging activity of the samples was significantly lower than the DPPH radical scavenging activity. This finding may be due to the difference in scavenging reactions that occurred in the aqueous phase of ABTS and the organic phase of DPPH [46].

#### 3.6.3. Reducing Power

Figure 4D shows the reducing capacity of the CPs. It can be seen from the figure that the reducing power of the sample increased with the increase of the sample concentration (*p* < 0.05). The reducing power of 4FT-CP (A_700nm_ = 0.071) was significantly lower than those of the other CP samples when the concentration was at 1.5 mg/mL, while the value of the 4FT-CP (A_700nm_ = 0.061) was slightly higher than those for the other three CP samples (A_700nm_ = 0.055–0.057) at a concentration of 1 mg/mL. In general, all of these CP samples exhibited a remarkable ability to capture radicals (DPPH^+^, ABTS^+^) and reduce ferric to ferrous ions.

## 4. Conclusions

In the present study, we explored the impact of different processes for the extraction of phycocyanin from the dry biomass of *Arthrospira platensis* on the physicochemical properties and antioxidant activities of CPs. According to the results of this study, the freezing and thawing technique, combined with ultrasonication, showed the best performance than the other three methods in terms of yield (15.92%) and average particle size (719.1 nm), while the high-pressure cell disruption process was better than the others in obtaining a product with a higher CP concentration (15.3%), purity (1.66 ± 0.16), *ζ* potential (−38.8 mV), DPPH (IC_50_ = 0.68 mg/mL), and ABTS (IC_50_ = 1.28 mg/mL) radical scavenging activity. Phycocyanins extracted by different methods have different secondary and tertiary structures. In all the methods, the high-pressure cell disruption process could be used as an effective method to obtain food-grade C-phycocyanin. The understanding of the effects of extraction methods on their properties could assist in selecting the appropriate extraction method to optimize the utilization of CP fraction as an alternate functional food for the food and drug industry.

## Figures and Tables

**Figure 1 foods-11-01296-f001:**
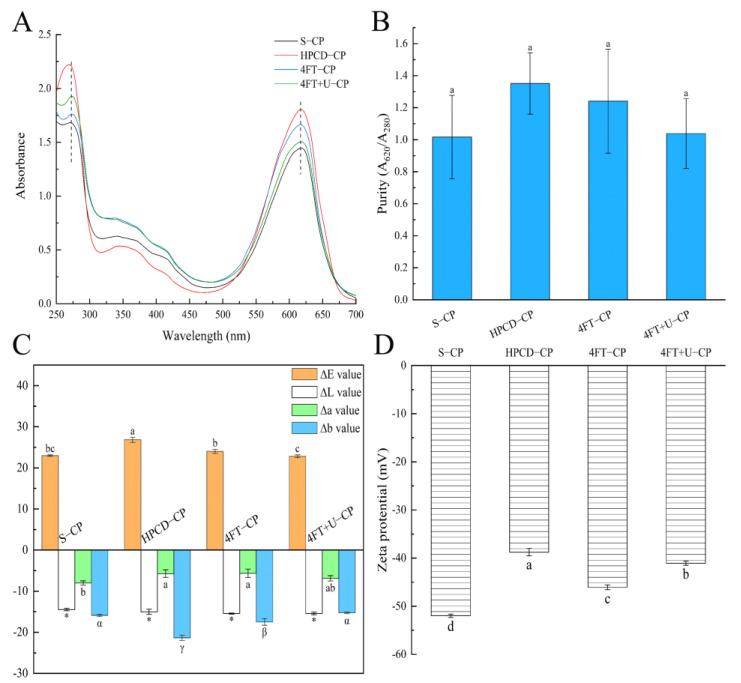
Absorption spectra (**A**), CP purities (**B**), color analysis (**C**), and zeta potential (**D**) of CPs obtained by different extraction methods. Different lowercase letters or single asterisks indicate significant differences (*p* < 0.05).

**Figure 2 foods-11-01296-f002:**
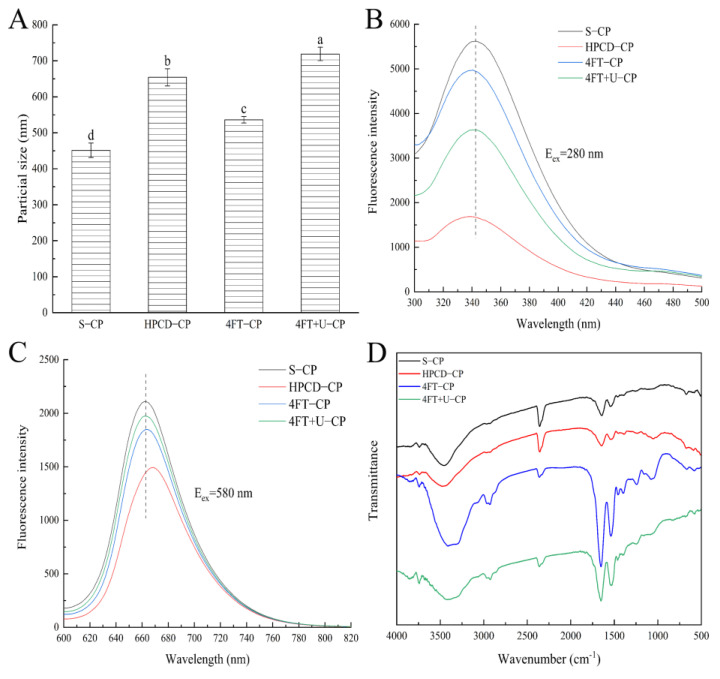
Average particle size (**A**), emission fluorescence spectra (using two different excitation wavelengths: 280 nm (**B**) and 580 nm (**C**)), and FTIR spectra (**D**) of CPs obtained using the different methods. Different lowercase letters indicate significant differences (*p* < 0.05).

**Figure 3 foods-11-01296-f003:**
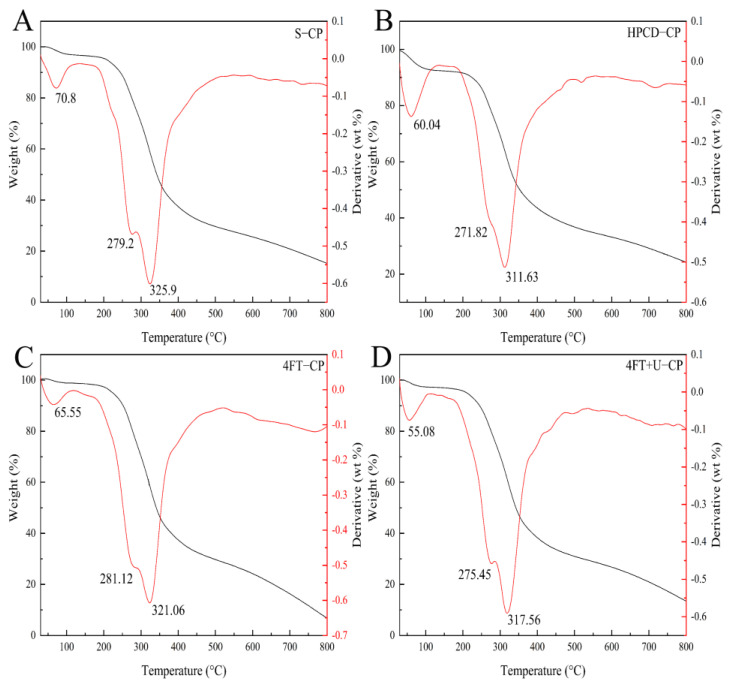
TGA of S-CP (**A**), HPCD-CP (**B**), 4FT-CP (**C**), and 4FT+U-CP (**D**) (protein concentration used was 1 mg/mL).

**Figure 4 foods-11-01296-f004:**
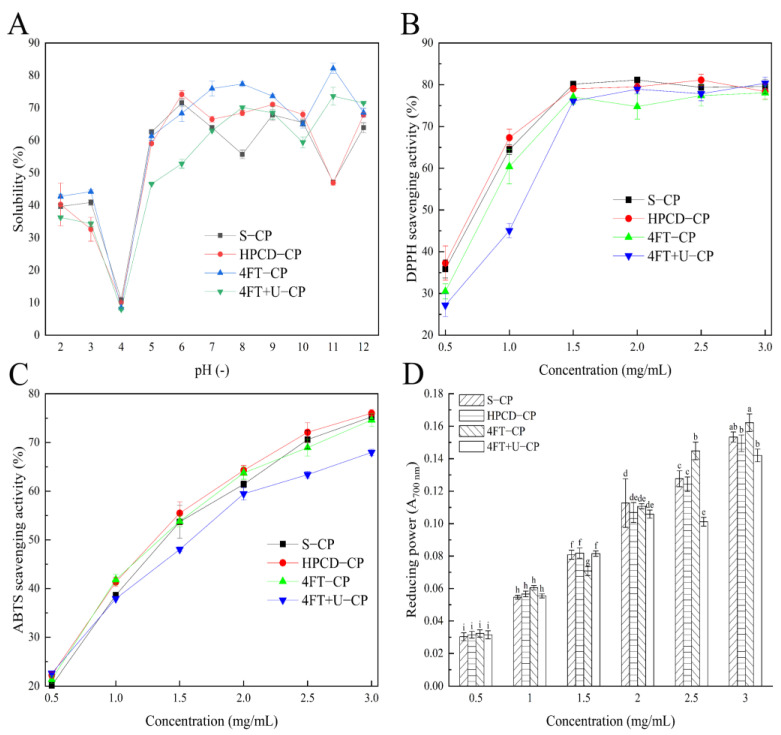
Solubility (**A**), free radical scavenging activities, and total reducing power of CPs obtained using different methods: DPPH radical scavenging activity (**B**), ABTS radical scavenging activity (**C**), and total reducing power (**D**). Different lowercase letters indicate significant differences (*p* < 0.05).

**Table 1 foods-11-01296-t001:** The CP content, protein content, and yield of CPs obtained by different extraction methods in wet mass.

	CP Content (wt%)	Protein Content (wt%)	Yield (%)
S-CP	10.9 ± 0.00 ^c^	79.1 ± 0.7 ^b^	15.9
HPCD-CP	15.3 ± 0.00 ^a^	77.8 ± 0.2 ^b^	5.8
4FT-CP	13.6 ± 0.53 ^b^	83.7 ± 2.2 ^a^	9.8
4FT+U-CP	12.2 ± 0.45 ^c^	81.1 ± 0.5 ^ab^	15.92

Different superscript letters in the same column indicate significant differences (*p* < 0.05).

**Table 2 foods-11-01296-t002:** Estimation of secondary structure in amide I region of CPs obtained by the different methods.

Sample	Area (%)			
	α-Helix	β-Sheet	β-Turn	Random Coil
	1660–1650 cm^−1^	1640–1600 cm^−1^	1670–1660 cm^−1^	1650–1640 cm^−1^
		1690–1670 cm^−1^	1700–1690 cm^−1^	
S-CP	19.16	54.14	26.70	0
HPCD-CP	0	55.46	22.08	22.46
4FT-CP	28.31	49.50	22.19	0
4FT+U-CP	28.66	49.84	21.50	0

## Data Availability

Data is contained within the article and available on request from the corresponding author.

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
