# Peer review of "Effect of Different Extraction Methods on Physicochemical Characteristics and Antioxidant Activity of C-Phycocyanin from Dry Biomass of Arthrospira platensis"

_foods, 2022, doi:10.3390/foods11091296_

Round 1

Reviewer 1 Report

The article titled: "Effect of different extraction methods on physicochemical characteristics and antioxidation of C-Phycocyanin from dry biomass of Arthospira platensis " could be interested for Foods readers, but in present form it cannot be published. The article needs major revision, according to comments and recommendations given below.

  1. lines 15-16 - the sentence is not clear; please rearrange it.
  2. line 45 - what does it mean: "and so on" - it is not a scientific language - please rewrite the sentence.
  3. line 48 - PEF showed higher purity (of what?) - the sentence is not clear and should be corrected.
  4. line 56 - it should be: "functional and physicochemical properties"
  5. line 79 - what do the Authors mean by: "trice"? (three times?)
  6. line 82 - the subtitle should be more detailed - please change it.
  7. line 104 - please explain, why such parameters for US were chosen in the study.
  8. line 113 -how do the Authors know/check, that the precipitated proteins contained mainly C-PC?
  9. line 170 - "the solution containing DPPH solution" - the style is incorrect.
  10. line 211 - it should be: "the shortest"
  11. line 255 - it should be: "might be related to the fact, that there are..."
  12. lines 254-257 - please give literature citation to this statement.
  13. lines 278-280 - please explain this phenomenon, it is not clear based on the given information.
  14. line 281  - please change the subtitle to more detailed.
  15. lines 282-285 - the sentence is incorrect, please rewrite it.
  16. line 286 - the second stage should begin at 150C, not at 50C.
  17. lines 294-296 - it is not clear, please give more details.
  18. lines 300-304 - the explanation is not clear, please rearrange it.
  19. lines 315-317 - is the high solubility enough evidence to conclude, that C-PCs with different extraction method may be applicable to food processing - please explain.
  20. line 329 - "different C-PCs from different ..." - the style is not correct.
  21. lines 357-358 - it is not a sentence - please correct.
  22. lines 367-368 - there were not enough researches done to conclude about sthe structures.
  23. line 368 - it can not be said that it is a new method to obtain food grade C-phycocyanin.
  24. Conclusions should be rearranged to present the most important information from the study.

Reviewer 2 Report

The paper "Effect of different extraction methods on physicochemical characteristics and anti oxidation of C-Phycocyanin from dry biomass of Arthospira Platensis" focuses on the application of different technologies for the extraction of phycocyanin from Arthrospira platensis cyanobacterium. The extracts obtained from the different technologies applied are characterized for their antioxidant ability and physicochemical properties. However, the manuscript presents several critical problems:

  • the authors stated to perform a comparison among the different method tested, nevertheless, there is only a partial explanation about the reason why specific parameters have been adopted during the extraction processes; yet, no optimisation has been performed or references provided for some of the technologies tested, thus this type of information should be provided if a comparison has to be performed;
  • The discussion of the results is, in some sections, poorly made e.g. line 219: the deltaL of HPCD sample is stated as the highest, however, telling by the figure provided the value is very similar to all the others with no statistical difference as well as the purity and this is not mentioned at all
  • The paper is characterised by many flaws in terms of spelling of the terms used e.g. the cyanobacterium Arthrospira platensis is spelled in different wrongly ways (including the title where the "r" is missing) throughout the manuscript. Yet, the term antioxidation should not be used as the property is usually referred as antioxidant ability/capacity. Moreover, some figures have different type of formatting styles (see Fig. 1 where some letters are bold style and others are not)
  • Above all the main problem of the manuscript is the English language quality which is very poor and it makes even hard to clearly review the paper

Reviewer 3 Report

This paper is about extraction methods but information about the production of dry biomass of Arthospira platensis should be given as this may affect the extraction, which will be useful in future works.

The manuscript is very hard to read due to the abusive use of acronyms. Since Chloro-Phycocyanins (C-PCs) are the objective of extraction, the authors do not constantly repeat the C-PC. Also, instead of CP-C, CP could be used (is simpler).

Line 61                 What is ultrasonication at 40% ?

L71 and 72          Correct big to small case on Boiling, Ammonium and Sulfate. Similar errors appear along the rest of the text.

Line 83                 The word wights is used often. Do you mean weights?

Line 119               Using a standard A4 paper is not a color standard. Is there a reasonable explanation why you did not use a calibrated color standard?

Line 158               Delete “degree”.

equation 5, 6, 7                 multiply by 100

Line 200               For better reading, the text could be rewritten like this: Table 1 shows the chloro-phycocyanins content, its yield and the protein content. The CP yields were 15.90% for the swelling method, 5.80% for the high-pressure, 9.80% for the freeze-thawing method and 15.92% for the  ultrasonication with freezing and thawing method.

Table 1                 Indicate that the contents are in dry or wet mass.

Line 210               review the English

Line 284               Pires and Soldi?! What is this?

Line 300               Delete “the” from the beginning of the sentence and rephrase.

Line 303               How the amount the protein influences the temperature?

Figure 4               Figure 4d is unreadable. Improve it. Make all 4 figures uniform.

Line 327               Why slightly lower?

Table 2                 Delete table. results are already in figure 4.

Line 338               Why it is obvious?

Round 2

Reviewer 1 Report

Authors have answered most of the comments and recommendations of the reviewer. The article still needs some major revision in English grammar and style, the time of the sentences should be constant, for example the sentence: "The ζ potential of HPCD-CP (-38.8 mV) was highest, but the average particle size of 4FT+U -CP (719.1 nm) was highest" is still not correct and should be rewritten. I suggest that the article should be corrected by native speaker.

Author Response

The full text has been revised using the English language service.

Reviewer 2 Report

The manuscript quality has been improved but still some parts remain unclear because of poor english language, some suggestions below but I strongly advice to ask support for a language editor 

in details:

-line 37: Arthrospira is spelled wrongly, please correct

-line 57: put ":" after extraction methods

-line 87: the word "dissociated" sounds inappropriate in this context, please replace with a most appropriate one

-Equation 1 is reported wrongly, should be A652 instead of A650

-Lines 212-215: the two sentences are not clear and difficult to understand, please revise it

- Figure 2 caption: please use "absorption spectra" since you have more than one instead of spectrum

-Figure 2 caption: three different excitation wavelengths are indicated but only 2 reported, please revise

-line 273: should be "the largest"

-line 294: "Figure 3 shows that" please add'

-line 295-296: sentence is not clear, please revise it

-line 347: "When the concentration..." apparently the verb is missing , please revise or improve it

-line 371-372: please improve the sentence because is not clear

-line 378: please add a comma after "While"

-line 382-386: sentences are not clear and need improvement, please revise the quality of the english for entire paragraph or it will be misunderstood

Author Response

The manuscript quality has been improved but still some parts remain unclear because of poor English language, some suggestions below but I strongly advice to ask support for a language editor.

Response: Thanks, the full text has been revised using the English language service.

in details:

-line 37: Arthrospira is spelled wrongly, please correct

Response: The part mentioned here has been revised.

-line 57: put ":" after extraction methods

Response: The part mentioned here has been revised.

-line 87: the word "dissociated" sounds inappropriate in this context, please replace with a most appropriate one

Response: We have changed “dissociated” to “separated”.

-Equation 1 is reported wrongly, should be A652 instead of A650

Response: The part mentioned here has been revised.

-Lines 212-215: the two sentences are not clear and difficult to understand, please revise it

Response: The part mentioned here has been revised, here has a mistake, now the sentences read: “This was consistent with the study of Tavanandi et al. [21], as ultrasonication presented stronger to break down the Arthrospira platensis cell walls [16]. During ultrasonication, intense local shock waves corresponding to thousands of atmosphere pressure were produced to destroy the cell walls. In Figure 2(a), three peaks at 350 nm, 375 nm, and 424 nm, also present in the spectra of ultrasonicated sample. This is because more other compositions were obtained in crude 4FT+U-CP.”

- Figure 2 caption: please use "absorption spectra" since you have more than one instead of spectrum

Response: Thank you very much for your careful reading. The part mentioned here has been revised.

-Figure 2 caption: three different excitation wavelengths are indicated but only 2 reported, please revise

Response: Thank you very much for your careful reading. The part mentioned here has been revised.

-line 273: should be "the largest"

Response: The part mentioned here has been revised.

-line 294: "Figure 3 shows that" please add'

Response: The part mentioned here has been revised.

-line 295-296: sentence is not clear, please revise it

Response: The part mentioned here has been revised.

-line 347: "When the concentration..." apparently the verb is missing, please revise or improve it

Response: The part mentioned here has been revised.

-line 371-372: please improve the sentence because is not clear

Response: The part mentioned here has been revised, now the sentence read: The reducing power of 4FT-CP (A700 nm=0.071) was significantly lower than those of the other CP samples when the concentration is at 1.5 mg/mL, while the value of the 4FT-CP (A700 nm=0.061) was slightly higher than those for the other three CP samples (A700 nm=0.055-0.057) at the concentration of 1 mg/mL.

-line 378: please add a comma after "While"

Response: The part mentioned here has been revised.

-line 382-386: sentences are not clear and need improvement, please revise the quality of the english for entire paragraph or it will be misunderstood

Response: The part mentioned here has been revised, now the sentence read: In all methods, high pressure cell disruption process could be used as an effective method to obtain food grade C-phycocyanin. The understanding of effects of extraction methods on their properties could assist in selecting the appropriate extraction method to optimize the utilization of CP fraction as an alternate functional food for food and drug industry.

Reviewer 3 Report

Overall:

This paper still deserves improvement of the English.

Change CPs to CP.

Line 13                 Change C-CP to CP.

Line 84                  Revise English.

Line 99                 Revise English.

Line 128               Not clear. Is the color being measured in a solution oro n the solid biomass?

Line 218               Delete “during high pressure cell disruption”.

Line 319              Do you mean: “application of mechanical forces, in the case of high-pressure cell disruption and ultrasound assisted with freezing and thawing process”?

Line 323               Are instead of were.

Author Response

Overall: This paper still deserves improvement of the English. Change CPs to CP.

Response: The full text has been revised using the English language service. And all the CPs need change to CP have been revised.

Line 13: Change C-CP to CP.

Response: The part mentioned here has been revised.

Line 84: Revise English.

Response: Thank you very much for your careful reading. We have changed this sentence to “The swelling extraction process was performed based on the method described by Li et al. [16] with slight modification.

Line 99: Revise English.

Response: The part mentioned here has been revised. We have changed this sentence to “Freezing and thawing extraction process was accomplished according to Antecka et al. [20]

Line 128: Not clear. Is the color being measured in a solution or on the solid biomass?

Response: Thank you very much for your careful reading. The part mentioned here has been revised. The color was measured in a solution.

Line 218: Delete “during high pressure cell disruption”.

Response: The part mentioned here has been revised.

Line 319: Do you mean: “application of mechanical forces, in the case of high-pressure cell disruption and ultrasound assisted with freezing and thawing process”?

Response: That's exactly what we're saying.

Line 323: Are instead of were.

Response: The part mentioned here has been revised.